# HIV-Associated Neurocognitive Disorder: A Look into Cellular and Molecular Pathology

**DOI:** 10.3390/ijms25094697

**Published:** 2024-04-25

**Authors:** Landon John-Patrick Thompson, Jessica Genovese, Zhenzi Hong, Meera Vir Singh, Vir Bahadur Singh

**Affiliations:** 1Department of Life Sciences, Albany College of Pharmacy and Health Sciences, Albany, NY 12208, USA; 2Department of Neurology, University of Rochester, Rochester, NY 14642, USA

**Keywords:** HIV, HAND, neuroinflammation, microglia, latency, platelet

## Abstract

Despite combined antiretroviral therapy (cART) limiting HIV replication to undetectable levels in the blood, people living with HIV continue to experience HIV-associated neurocognitive disorder (HAND). HAND is associated with neurocognitive impairment, including motor impairment, and memory loss. HIV has been detected in the brain within 8 days of estimated exposure and the mechanisms for this early entry are being actively studied. Once having entered into the central nervous system (CNS), HIV degrades the blood–brain barrier through the production of its gp120 and Tat proteins. These proteins are directly toxic to endothelial cells and neurons, and propagate inflammatory cytokines by the activation of immune cells and dysregulation of tight junction proteins. The BBB breakdown is associated with the progression of neurocognitive disease. One of the main hurdles for treatment for HAND is the latent pool of cells, which are insensitive to cART and prolong inflammation by harboring the provirus in long-lived cells that can reactivate, causing damage. Multiple strategies are being studied to combat the latent pool and HAND; however, clinically, these approaches have been insufficient and require further revisions. The goal of this paper is to aggregate the known mechanisms and challenges associated with HAND.

## 1. Introduction

Although combined antiretroviral therapy (cART) has made HIV a manageable and treatable disease, resulting in similar life expectancy for people living with HIV (PLWH) compared to their non-infected counterparts, HIV-associated neurocognitive disorder (HAND) remains a significant challenge [1,2]. For PLWH, the estimated percentage of those experiencing HAND is around 50%, with marginal differences observed across the United States, Europe, Africa, and Asia [3]. Clinical manifestations of HAND have been divided into three severities, asymptomatic neurocognitive impairment (ANI), mild neurocognitive impairment (MNI), and HIV-associated dementia (HAD). ANI is characterized by motor, memory, and executive functioning loss not impacting the daily life of those living with it. MNI is slightly more severe and is defined by impairments beginning to impact daily life. Lastly, HAD is the most severe and is associated with severe motor and memory loss, which usually leads to death within one year [4]. At a cellular level, the interactions that lead to the development and progression of HAND are complex and include both host and viral factors that are at the forefront of HIV research today. From early invasion into the brain to chronic inflammatory states lasting decades, many groups have elucidated various mechanisms associated with HAND. The goal of this paper is to describe the mechanisms associated with the onset and progression of HAND and discuss current management strategies.

## 2. Early Infiltration of HIV to the Central Nervous System

A 1992 case involving a blood transfusion contaminated with HIV led to the discovery that HIV enters the CNS within 15 days of exposure [5]. At this time, it was unknown how HIV entered the immune-privileged brain, especially early into infection. One of the prevailing theories was the “Trojan Horse” model. In this model, infected monocytes would be able to cross the blood–brain barrier (BBB) to gain access early in infection. By 2006, this theory garnered strong evidence showing that HIV-infected monocytes can cross the BBB through the expression of C-C motif receptor 2 (CCR2) on infected monocytes, which was attracted to secreted C-C motif ligand 2 (CCL2) [6]. This was further supported in 2010 by showing that HIV-infected monocytes expressed elevated levels of CCR2 and were more responsive to CCL2 [7]. In 2012, a study in Thailand demonstrated that HIV mRNA is detectable in the CNS within 8 days of suspected exposure [8]. This study sparked the interest of many in the field and has led to the development of several models of HIV to bypass the BBB to gain entry to the CNS. Recently, it has been shown that HIV can infect pericytes in the BBB [9]. The infection of pericytes can allow HIV to be secreted into the CNS where it can infect the resident microglia. This would provide a direct route for HIV to enter the brain without needing to hijack immune cells in order to bypass the BBB for entry.

Of interest to our group is the role of platelet activation and platelet leukocyte complexes (PLCs) in the infiltration of HIV into the brain. We showed that during HIV infection, there is an activation of platelets to secrete CD40 Ligand (sCD40L) [10]. Subsequently, we demonstrated that the sCD40L promotes the development of platelet monocyte complexes (PMCs) during HIV infection through the interaction of P-selectin on platelets and the P-selectin glycoprotein ligand-1 (PSGL-1) on monocytes [11]. Interestingly, the most increased phenotype of monocytes in the PMCs we found was CD16+ monocytes [10]. CD16+ monocytes display a more inflammatory phenotype than CD16− monocytes, prompting the notion that these circulating complexes are the most significant PLC for promoting chronic inflammation and it had been previously shown that during HIV infection, the vast majority of monocyte-derived immune cells that enter the brain are CD16+ [12,13]. These previous studies paired with our work show that PMCs, particularly CD16+, are significant in promoting the transmigration of inflammatory monocytes into the brain. We are currently analyzing different strategies to antagonize these complexes to reduce further transmigration to the brain, as well as reduce persistent inflammation associated with these cell complexes.

## 3. Loss of Blood–Brain Barrier Integrity

The deterioration of the BBB has been shown to progress to neurocognitive dysfunctions and dementia [14,15]. By 2002, it had been shown that breakdown of the BBB was one of the key markers of HAD [16]. The mechanisms behind HIV-associated BBB breakdown represent an interesting interplay between host factors and viral proteins. One of the first pathways characterized by the HIV-related dysregulation of the BBB was trans-activator of transcription (Tat) toxicity on brain microvascular endothelial cells (BMECs) [17]. Tat’s primary function for HIV is to promote the transcription of long viral transcripts for the integrated provirus. It achieves this by binding to host factors in the P-TEFb complex and TAR region in short viral transcripts to cause the continual activation of CDK9 in the P-TEFb complex. This results in the expression of viral transcripts long enough to encode key proteins [18,19,20]. In Tat-mediated BMEC toxicity, Tat is able to induce the activation of the antioxidant nuclear factor-кB (NF-кB) and activator protein-1 (AP-1), which damage the BMEC and promote monocyte chemoattractant protein, which further propagates inflammation by recruiting circulating monocytes [17]. Tat-mediated toxicity has been one of our group’s interests for many years and we have demonstrated an interesting interplay between Tat and sCD40L. We have shown that Tat is able to upregulate CD40 in monocytes and microglia through the activation of NF-кB. We also have shown that Tat, in combination with sCD40L, exacerbates the activation of NF-кB signaling and the release of tumor necrosis factor-alpha (TNF-α) [21]. Our results show that Tat alone is able to induce host factors that can further amplify the inflammation [21].

In addition to attracting and activating immune cells that propagate an inflammatory response, Tat also damages the BBB by weakening tight junctions and deregulating cellular migration through the BMEC layer. Specifically, Tat acts by downregulating the transcription of the tight junction proteins occludin and zonula occluden 1 and 2 (ZO-1 and ZO-2), while also promoting the accumulation of reactive oxygen species (ROS), advanced glycation end products (AGEs), and amyloid beta (Aβ) in the brain [22,23,24]. The downregulation of the tight junction proteins is associated with interference of Ras signaling through the stimulation of the Ras homolog gene family A (RhoA)/Rho-associated kinase (ROCK) pathway. The accumulation of ROS is associated with the activation of the Ras pathways stimulated, while the buildup of Aβ and AGE has been shown to be caused by the Tat-mediated upregulation of the Aβ transfer receptor lipoprotein receptor protein 1 (LRP-1), and the receptor of AGE (RAGE) [23,24]. Another pathway our group elucidated for the Tat-mediated disruption of tight junction proteins is through the inhibition of Sonic hedgehog (Shh) signaling. Using a transgenic Tat-expressing mouse model, we demonstrated that Tat expression decreased protein levels of Shh, Gli 1 (a key transcription factor for tight junction proteins), and the tight junction protein Claudin5 [25]. In addition, we also used a humanized mouse model to show that during HIV infection, a mimetic of Shh, the smoothened agonist (SAG), can prevent the downregulation of Gli1, occludin, and Claudin5 while also protecting from leukocyte invasion in acute and chronic phases of infection [26,27]. Together, our work and others have shown the different ways that Tat is involved in disrupting tight junctions and the overall disruption of the BBB during HIV infection.

Another key HIV protein that damages the BBB is gp120. gp120 had been shown to be secreted at low levels in the blood, although due to technical limitations, it was not shown until 1999 that gp120 is secreted at low levels in the brain as well [28]. This led to the question of whether gp120 alone is significant enough to cause neuronal cell death and/or the breakdown of the BBB. gp120 was first shown to be important to HAND by showing its toxicity to neurons by inducing apoptosis by the activation of the CXCR4 receptor on neurons [29]. After the direct pathway for neuronal cell death was characterized, further investigation with mouse models and macrophage co-culture systems demonstrated that neuronal cell death is predominately associated with the activation of macrophages and cell-to-cell signaling and the p38 MPAK cell death pathway [30]. Shortly after these findings, it was shown that gp160 and gp120 could induce apoptosis in endothelial cell lines as well [31]. Lastly, gp120 alone was shown to weaken BBB tight junctions and allow for elevated levels of monocyte transmigration [32]. This suggests that not only does gp120 have a role in the long-lasting BBB weakening, but it is also involved in early damage and immune cell invasion.

## 4. Activation of Immune Cells Exacerbates Neuroinflammation

### 4.1. Microglial Targeting of Synapses

The activation of microglia significantly contributes to HAND progression. Microglia of the CNS are essential to maintain the overall homeostasis of the brain. They are involved in critical functions that regulate brain development, neuroplasticity, and neuron injury repair [33]. These specialized cells become activated when small changes in the CNS are detected [34]. The dysfunction and overactivation of microglial cells are implicated in several neurodegenerative diseases.

A necessary receptor that is crucial for the survival and function of microglia is the colony-stimulating factor receptor (CSF1R) [35]. Previous studies using mouse models have shown that the inhibition of this receptor results in approximately 99% of brain-wide microglia depletion [36]. When activated, this receptor can regulate two microglial phenotypes: the M1 phenotype is pro-inflammatory and associated with neurotoxicity, while the M2 phenotype is anti-inflammatory and associated with neuroprotection [36,37]. Microglia’s role within the CNS is tightly regulated by “on” and “off” signals that create a reactive/phagocytic or a quiescent state, respectively. These signals are released by neurons and astrocytes through the TGFβ2 signaling pathway [38,39,40].

Upon stimulus or injury, pro-inflammatory signals from microglia are transduced throughout the cellular network of resident CNS immune cells. As a result, inflammation occurs in localized areas of the brain, resulting in the upregulation of phagocytosis, cytokine secretion, inflammasome activity, and immune cell proliferation [41]. During HIV infection, Tat can induce a shift to the M1 phenotype [42]. While this is thought to be one of the reasons for prolonged damage of the CNS seen in HAND, M1 microglia stimulated with TNF-α and IFN-γ have been shown to contain HIV replication pre- and post-integration by the upregulation of APOBEC 3A [43]. This creates a challenging paradigm for preventing inflammation, as M1 activation is inherently leading to inflammatory cytokines, but also prevents inflammation by restricting HIV replication. This restriction has also been linked to inducing latency in microglia, which further complicates the picture for developing treatment strategies [44].

In part, infected microglia exert their effects by directly targeting the neuronal synapses. In a developing brain, these cells are involved in the establishment of neural circuits through synaptic pruning. However, dysregulated microglia can cause abnormal pruning that results in the loss of synaptic connections and degeneracy [45]. Several studies have provided mechanisms that suggest an interaction between virally induced/activated microglia and synaptic proteins that cause phagocytosis of the synapse.

Since the 1990s, neuron loss and cortical brain damage have been reported in PLWH. In such cases, correlations were observed between dysregulated fractalkine microglial receptors (CX3CR1/FKN) and synapse loss. A recent study investigating HIV-associated pain revealed a possible connection between synaptic degradation and the gp120-induced upregulation of both CX3CR1 and CX3CL1, a chemoattractant typically secreted by neurons during toxic insults. When treated with gp120 alone, an increase in CX3CR1 activity in microglia was seen along with a significant decrease in three markers of presynaptic health in both neuron–glial co-cultures and mouse models: Syn I, Syt-1, and PSD-95 [46].

The coupled response between increased glial receptors and increased neuronal cytokine expression perfectly positions microglia to target the synapse for phagocytosis. With regards to the CNS and PLWH, the ramifications of induced microglia could include the extensive elimination of neuronal circuits. This is especially concerning as the continuous activation of microglia due to HIV latency would only promote further neurodegeneration.

### 4.2. Recruitment of Circulating Monocytes

Circulating monocytes are recruited to the CNS by secreted CCL2. As with synaptic degradation, HIV viral protein gp120 is implicated in altering the activity of resident CNS immune cells [6]. This includes both microglia and astrocytes through an increased CCL2 expression upon gp120 treatment. As a result, monocytes are recruited to where they are needed, expand in population, and provide an immune response. The exposure to CCL2 upregulates the monocyte expression of adhesion molecules. Because of this, gp120 might induce monocyte interaction with endothelial cells and platelets [47].

The transudative mechanism of peripheral monocytes through endothelial cells (ECs) into subendothelial spaces allows monocytes to reach targeted locations. During an immune response, the activation of pro-inflammatory transcription factors like NF-кB has been shown to upregulate cell surface receptors collectively known as cellular adhesion molecules (CAMs/ICAMSs) [47,48]. It has been recognized for some time that HIV infection leads to the increased activation of NF-кB (subsequent increase in CAMs/ICAMS). More recent studies have shown that in addition to gp120, Tat also induces the expression of these adhesion molecules [49].

The upregulation of these cell surface receptors allows for increased interactions between peripheral monocytes and endothelial cells of the BBB. Because of this, monocytes are encouraged to migrate across the BBB and invade the CNS. As previously mentioned, activated monocytes form a complex with activated platelets, known as a platelet–monocyte complex (PMC). The formation of PMCs in addition to increased CAM/ICAM expression on both endothelial and monocytic cells further enhances the migration of monocytes across the BBB. This persistent cellular movement sustains the immune response and propagates pro-inflammatory signals within the CNS as invading immune cells are activated. Ultimately, in an attempt to localize and combat viral infection, the early activation of microglial and then peripheral monocytes exposes the double-edged sword of the host’s immune response.

### 4.3. Other Long-Lasting Cytokine Pathways

There are several cytokine and chemokine cues from the environment that are implicated in the persistent neuroinflammation of neurodegenerative diseases. Direct evidence for this was produced in 2005 when Cartier et al. showed elevated levels of pro-inflammatory and apoptotic molecules (IL-1α, CXCR2, CCR3, CCR5, and TGF-β) in post-mortem brain tissue in subjects with Alzheimer’s disease (AD) compared to age-matched controls [50]. Like other neurodegenerative diseases, the prognosis of HIV, a life-long viral infection, is intimately mediated by long-lasting dysregulated cytokine pathways.

Although the cytotoxic effects of Tat have been known for decades, in 2019, a strong connection was made between the downregulation of nod-like receptor CARD domain 5 (NLRC5) and NF-кB. Nod-like receptors are located on several glial cells and are dedicated to responding to pathogen-associated molecular patterns. In vitro exposure to Tat revealed a significant decrease in NLRC5 expression in the prefrontal cortex of mouse microglial cells [51]. NLRC5 is thought to downregulate the pro-inflammatory effects of NF-кB by blocking the phosphorylation of IKKα and IKKβ [52]. This is an important immunological break in the inflammatory cascade as the activation of NF-кB leads to inflammasome activity and inflammatory molecules. However, with Tat’s ability to decrease NLRC5, this break is diminished and the host immune response is kept on.

The well-known immunological signaling pathway JAK-STAT is critical for the coordination of immune cells and is implicated in HIV neuropathology. These interactions are bolstered as the downstream cytokines of JAK-STAT travel through the CNS in cerebrospinal fluid to interact with immune cells deep within cervical lymph nodes [53]. In 2018, it was demonstrated that HIV-1 viral protein Vif associates with STAT1 and STAT3, leading to their degradation through ubiquitination, as opposed to activation through phosphorylation. Without STAT1 and STAT3, T cells fail to receive Type 1 interferon (IFN) survival signals, resulting in their depletion [54].

Over the past decade, however, there has been growing evidence that disruption within this pathway may affect HIV latency. Due to insufficient activation signals from JAK-STAT, there is a possibility that memory T cells may be blocked from becoming a latency reservoir. Studies have shown a marked reduction in the number of CD4+ T cells harboring integrated HIV DNA in both in vitro and in vivo experiments using FDA-approved Jak inhibitors tofacitinib and ruxolitinib [55]. The lack of viral integration within these cells further implicates JAK-STAT’s involvement with HIV latency.

The mechanism behind both of these pathways creates a situation for the host in which early viral proteins elicit an initial host response that persists and becomes chaotic, distorted, and harmful to the individual. Further research into identifying the roles of the NLRC5-NF-κB signaling axis and the JAK-STAT pathway during an HIV infection is needed. Nevertheless, preliminary data provide a possible insight into the sustained neuroinflammation seen in PLWH. This information may be used in the future to develop novel treatments for HAND [51,55].

### 4.4. Biomarkers of HAND and Neurocognitive Impairment

There are two categories of biomarkers being investigated in the context of HAND, advanced neuroimaging markers and plasma/serum and CSF-based soluble markers. Among the neuroimaging markers, cerebral blood flow (CBF) and cerebrovascular reactivity (CVR) are markers for BBB dysfunction. Ances et al. have shown reduced resting CBF in the basal ganglia and visual cortex of PWLH compared to HIV-negative healthy controls (HCs) [56]. CBF abnormalities have also been documented in PWLH with cognitive impairment [56,57,58]. Callen et al. used changes in CBF induced by intravenous acetazolamide and found decreased CVR in the frontal lobe and basal ganglia, while another study used transcranial Doppler ultrasound and breath-holding to assess mean flow velocity changes to infer CVR [59,60]. Higher CVR was associated with a higher score on the Montreal Cognitive Assessment. In corroboration with these reports, our group found that CVR was a more sensitive measure of pre-cART neurovascular damage as compared to CBF [61]. In addition to these, axonal and myelin changes, extracellular free water (an index of inflammation), quantitative susceptibility mapping (to measure iron dysmetabolism), and white matter hyperintensities are other biomarkers that are being explored by our group and others [61,62,63,64,65,66,67,68,69,70,71].

Many different soluble markers in plasma, serum, and CSF have been investigated for their association with HAND such as markers of neuronal injury, endothelial dysfunction, and monocyte activation. A neurofilament light (NFL) chain is an essential scaffolding protein of the neuronal cytoskeleton, which is released into CSF and peripheral circulation upon axonal injury. Phosphorylated Tau is considered a marker for neuronal and synaptic loss. Glial fibrillary acidic protein (GFAP) is associated with glial activation. Multiple studies have shown that PLWH exhibit increased levels of NFL in blood plasma and CSF, especially in cART-naïve individuals [72,73,74,75,76]. A few reports have also found pTau and GFAP levels to be increased in PLWH; however, these findings have not been consistent throughout the different studies [77,78,79,80]. Lastly, endothelial activation has been measured via levels of the intracellular adhesion molecule (ICAM) and vascular cellular adhesion molecule (VCAM), and monocyte activation has been measured using levels of circulating monocytes and sCD14, sCD163, neopterin, etc. ICAM and VCAM are found to be elevated even in well-controlled HIV infection [61,81,82,83,84]. A study by Shikuma et al. showed that a higher antiretroviral monocyte efficacy score was linked with better cognitive performance [85]. HIV DNA levels in monocytes and soluble CD14 were found to be associated with VCI in PLWH [86,87]. In addition, our group and others have shown that CD16+ monocyte levels are also associated with a worse cognitive outcome in PLWH [88,89]. In resource-limited settings where MRI facilities are not readily available, these blood biomarkers could be more easily implemented. However, well-designed, prospective, longitudinal studies are needed to overcome the variations found across different studies.

## 5. Propagation of HIV-Associated Neuroinflammation

### 5.1. FKN Signaling

CX3CL1, or fractalkine (FKN), exists as an EC membrane-bound protein that serves cell-to-cell adhesion. However, when bound to a monocyte receptor (CX3CR1), it doubles as an immune cell attractant, promoting the recruitment of WBCs. Additionally, CX3CL1-CX3CR signaling is required for immune cells to communicate. This interaction suggests that the activation of the FKN pathway during an HIV CNS invasion may encourage further immune cell invasion and inflammation.

A study in 1997 showed that CX3CL1 is more highly expressed in the CNS compared to the surrounding periphery [90]. Additionally, this expression is increased during viral infections, such as HIV. As previously mentioned, the interaction between HIV Tat and sCD40L is able to activate the NF-кB pathway [21]. A downstream effect of this transcription factor is the upregulation of CX3CL1 expression. In doing so, further recruitment of immune cells such as CD8 T cells occurs [91]. The implications of CD8 T cell involvement include the release of potent and possibly toxic granules, along with dysregulated interactions with CD4+ T cells. Furthermore, the secondary activation of CD8 T cells during latent HIV infection has been shown to result in poor viral control [91].

The effects of altered CX3CL1 expression have been documented in other inflammatory diseases like coronary artery disease (CAD) and kidney disease. When comparing CX3CL1 mRNAs within the T cells of atherosclerotic plaques and peripheral blood, it was found that plaques contained a higher concentration [92]. In another study with similar findings, CX3CL1 expression in renal biopsies of patients with acute glomerulonephritis was elevated compared to negative controls [93]. In addition to an HIV infection of the CNS, a variety of neurodegenerative diseases involve pro-inflammatory signals and immune cell communication. Current evidence supports the involvement of the CX3CL1-CX3CR signaling axis and shows its possible association with other pro-inflammatory pathologies. The long-term dysregulated CX3CL1-CX3CR signaling response from the host during an HIV invasion of the CNS implicates immune cell migration, persistent inflammation, and possibly latency. However, there are other immunological pathways that link these disorders.

### 5.2. TNF-Alpha Signaling

Another important and well-studied immune molecule is TNF-α. Upon infection, TNF-α is released by macrophages and monocytes to stimulate inflammatory pathways and oxidative stress. Our previous work has shown that Tat in combination with sCD40L can increase TNF-α secretion through the activation of NF-кB [21]. Because of this, the early Tat gene of HIV is linked to the exacerbation of pro-inflammatory pathways. Two additional and important signaling effects of TNF-α include apoptosis and T cell differentiation [94]. Across the literature, the dysregulation of inflammatory pathways has emerged as a common cause of neurodegenerative diseases.

Although vastly different diseases, the neurodegeneration seen in Parkinson’s Disease (PD) and amyotrophic lateral sclerosis (ALS) show similar molecular patterns to those with HIV. For patients with PD, elevated levels of TNF-α were found in serum samples. Perhaps more importantly though, elevated TNF-α was positively correlated with the severity of disease [95]. In patients with ALS, elevated levels of TNF-α were found, in association with gene mutations SOD1 and C9ORF72 [96].

Since the 21st century, there has also been evidence showing elevated TNF-α in subjects with a variety of psychiatric illnesses linked to neuroinflammation [97]. One such illness is schizophrenia, a complex psychiatric disorder marked by hallucinations and/or delusions. Not only was an association between elevated TNF-α serum levels and schizophrenia demonstrated, but a positive correlation between schizophrenic symptoms and TNF-α was found [38,98].

With regards to HIV in the CNS, the implications of dysregulated TNF-α involve a widespread inflammatory response that leads to tissue damage. A recent study investigating the mechanism behind HIV-induced TNF-α signaling in monocytes showed that HIV-1 gp120 produced increased levels of TNF-α in a dose-dependent manner [39]. Interestingly, this report also showed that even when viral entry was blocked, increased levels of TNF-α were still observed. This implies that viral entry is not a requirement to alter TNF-α expression.

The relationship between pro-inflammatory molecules and different neurodegenerative diseases represents immunological pathways that have gone awry. This dysfunction can be seen in their overlap from laboratory results such as serum biomarkers to clinical presentations like premature cognitive decline. The causes of PD, ALS, schizophrenia, and HAND arise from vastly different methods, but one of the underlying mechanisms heavily involved appears to include TNF-α. Further investigation into the pathogenic inflammatory effects of this molecule may provide insight into treatment and possibly a cure for these neurodegenerative diseases.

### 5.3. Cofactors of HAND Progression

A variety of cofactors have been identified to worsen HAND development. One major cofactor of interest is substance abuse. A 2008 study found that 71% of PLWH reported substance abuse while just 24% of PLWH seek treatment for substance abuse [99]. The interplay between HAND progression and substance abuse remains challenging to study; however, some reports have shed insight into this relationship. Studies using simian immunodeficiency virus models in macaques showed an increased influx of macrophages and viral load in the CNS with exposure to cocaine and methamphetamine [100,101]. Other studies have shown that HIV replication in microglia can be upregulated during substance abuse as a response to increased dopamine from drug use [102,103]. For opiates, reports have shown that morphine and Tat synergistically promote neuronal cell death [104,105]. Clinically, it is unclear how substance abuse can shape HAND progression; however, one study showed that a history of substance abuse did not increase the rate of neurocognitive impairments [106]. Noteworthy for this study is that the majority of substance abusers had over a year of abstinence prior to the start, so it is unclear how active drug use may contribute to HAND progression.

A second cofactor to HAND progression is co-infection with both chronic and acute pathogens. In particular, hepatitis C virus (HCV) has garnered attention due to exacerbating neurodegeneration in PLWH. For HIV+HCV+ individuals, the rate of neurocognitive impairments is estimated to be around 63% compared to just 10–49% for HCV+ and 50% for HIV+ [3,107,108]. HCV has been shown to infect and persist in microglia, which leads to a chronic stimulus and cytokine release [109]. Two of these cytokines include TNF-α and IL-1ß, which are key contributors to HAND [94,110,111]. Further investigations are needed to provide a more comprehensive understanding of the interaction of HCV and HIV co-infection, particularly during active HCV replication, and how it can lead to neurodegeneration [107]. Two other pathogens that have been documented to exacerbate HAND include the intracellular parasite *Leishmania* and the *Mycobacterium* bacteria. *L. donovani* infects dendritic cells and macrophages in the blood where it upregulates IL-4 and IL-6, and during the HIV co-infection of macrophages, this upregulation is enhanced [112]. The increase in IL-4 and IL-6 during *L. donovani* likely can damage the BBB and CNS, although more evidence is needed to confirm this. *M. tuberculosis*, like HIV, can infect macrophages for years and produce a chronic inflammatory state. The co-infection of *M. tuberculosis* with HIV in the context of AIDS progression has been well established; however, latent infection with both pathogens in the context of HAND remains less studied [113,114]. During active TB, an increased level of HIV is detectable in the blood and CSF [115,116]. In the context of HIV impacting *M. tuberculosis*, HIV targets CD4+ T cells that have been shown to neutralize *M. tuberculosis* by IFN-γ and IL-2 induction, which can worsen *M. tuberculosis* outcomes in patients [117]. For latent infections of HIV and *M. tuberculosis*, there has not been a detectable chronic increase in cytokines compared to HIV alone; however, there is an increase in T cell activation markers on both CD4+ and CD8+ T cells, namely CD38+ and HLA-DR expression [118]. HLA-DR and CD38+ expression on CD8+ T cells has been shown to be a marker for HAND, but it is not clear if it is directly causative of HAND [119]. Another possible way *M. tuberculosis* could contribute to HAND during more latent states could be intermittent reactivation, but further investigation is needed. Overall, more research is needed to determine the role of these cofactors and others in the progression of HAND.

## 6. Impact of Latency on Neuroinflammation

### 6.1. Latency in the Central Nervous System

One of the most challenging aspects of HAND is the latent pool of infected cells that persist during cART. The development of cellular, three-dimensional organoid, and small animal models of HIV-1 infection in microglia has greatly improved our molecular understanding of how intermittent emergence of HIV-1 from latency in microglial cells primarily contributes to both neuroinflammation in the CNS and the progression of HAND [40,120,121,122,123,124].

With an average age of 4.2 years, microglial cells’ unique characteristic of slow cell division allows the persistence of HIV-1 in the brain throughout the life of the patients compared to other potentially infected cells in the brain [125,126]. Microglia also serve as the main producer of pro-inflammatory cytokines and partake in phagocytosis when purines released in the forms of ATP and ADP from damaged neurons bind to microglia purinergic receptors P2RY12 [127]. During an active infection, neurodegeneration can result from the persistency of pro-inflammatory markers, such as NO, ROS, IL-1ß, TNF-α, and IL-6 via AP-1 (FosB), NF-кB (RelA/NF-KBI), and Stat1 signaling in microglial cells [128]. A direct association has also been drawn between the persistent upregulation of pro-inflammatory markers, including IL-8, IL-1ß, and TNF-α, and reactivation of latent HIV [129,130,131].

In a case–control comparison study, direct evidence of HIV-1 latency associated with neurodegenerative disease from 10 autopsies out of 32 HIV-seropositive (HIV+) patients showed non-detectable p24 in the brain with a high amount of viral DNA [132]. Under normal circumstances, the physiological homeostasis of the CNS causes microglia to return to their “resting” state by signaling factors such as FKN, CX3CL1, CD200, TGFbeta1/2, and Wnt [133,134]. However, when there is an imbalance of physiological homeostasis with the prolonged overproduction of pro-inflammatory cytokines such as TNF-α, IL-1ß, IL-6, IL-8, CCL2, CCL5, ROS, and reactive nitrogen species (RNS) triggered by persistent invading viruses, there is a consequence of the exacerbation of neuroinflammation and a further neurodegenerative effect [133,135].

The periodic emergence of HIV-1 from latency within microglial cells may further contribute to neurological complications via the reactivation of more microglial latent reservoirs and proviruses from the recognition of inflammatory cytokines, such as TNF-α and IL-1ß that are associated with HAND [133,136]. Furthermore, the abnormal activation of microglial cells occurs in response to both (1) neuronal damage that results from chronic neuroinflammation and (2) inflammatory cytokines that are produced by reactive astrocyte neurodegeneration [137]. TNF-α triggers the activation of microglial cells, leading to the excessive activation of microglia, which leads to the accumulation of pro-inflammatory cytokines. Meanwhile, TNF-α induces the translocation of the transcription factor NF-кB to the nucleus and upregulates many inflammatory cytokines in primary astrocytes, resulting in neuronal death [110]. Moreover, TNF-α induces the expression of FKN, which enhances the adhesion, chemoattraction, recruitment, and activation of other inflammatory cells [138,139,140]. Similarly, the expression of IL-1β from microglia is tightly regulated by caspase-1, whose activation is triggered by cleavages of procaspase-1. Once IL-1 is activated, IL-1β activates astrocytes, which subsequently produce TNF-α [139]. Studies have shown that gp120 upregulates both the expression of IL-1β and TNF-α, which indirectly activates latently infected microglial cells [141,142,143]. The cleavage of IL-1 to its active form, IL-1β, has been shown to be a result of Tat and gp120-mediated activation of the NLRP3 inflammasome [144,145]. This interplay between neuronal damage and microglia activation in the context of latency in the CNS and the progression of HAND continues to be an important area of in-depth research.

Astrocytes have also been shown to support latent HIV reservoirs in the CNS [146,147]. In the viral production in the CNS, astrocytes are typically less studied as they have been shown to support limited viral production. This limited production can be overcome by a stimulus from IFN-γ and TNF-α [148,149]. In the context of harboring a latent pool, it has been shown that class I histone deacetylases and histone methyltransferase, SU(VAR)3-9, repress viral production and promote latency in astrocytes [149]. It is unclear how significantly the astrocytic latent pool contributes to the progression of HAND and further research is needed to fully understand its impact.

### 6.2. The Circulating Latent Pool

Microglial cells are considered the most challenging HIV reservoir in CNS. Likewise, circulating T cells are characterized as the major HIV reservoir in the peripheral blood. Most studies have found that memory T cells are the majority of latently infected cells. During HIV entry into different T cell subsets, the support of proviral transcription, proviral repression, and latency would be determined by the different transcriptional and intrinsic cellular factors that the virus encountered. Some of the essential host transcription factors, such as NF-кB, AP-1, NFAT, Sp1, and processive RNAP II, can assist efficient proviral transcription once the viral genome is integrated [150,151,152]. On the other hand, studies have shown that three main biochemical mechanisms limit the proviral transcription: (i) absence of the positive transcription factors like NF-кB, (ii) epigenetic changes on chromatin, and (iii) the presence of repressive factors [153,154]. Latent HIV infection appears to be mostly contained within the central CD4+ T cell compartment with a minority contained in naïve T cells [155,156]. Two cellular mechanisms for the generation of latent infection in CD4+ T cells could be categorized as follows: (1) during the transition from activated CD4+ T cells to a resting state and (2) the direct infection of resting CD4+ T cells [157,158].

Both naïve and activated CD4+ T cells can be infected and support latency. Resting CD4+ T cells can be infected through cell–cell contact by taking viral particles from dendritic cells that present with Siglec-1/CD169 and DC-SIGN receptors. In HIV-1-infected resting CD4+ T cells, no infectious particles were produced. Instead, these cells were observed to have provirus-integrated near transcriptionally active chromatin, which allows the expression of some spliced HIV RNA producing viral proteins, such as Gag and Nef but no to negligible production of Tat, Rev, and Env [159,160,161]. On the other hand, activated CD4+ T cells contain crucial metabolites such as nucleotides and amino acids that are required for HIV replication and viral transcription. Activated CD4+ T cells will then establish and maintain latency by returning back into quiescent resting memory CD4+ T cells or harbor latent HIV provirus. The subsequent reactivation of quiescent resting memory CD4+ T cells will be triggered by the activation of TCR with the same antigen, leading to the TCR-induced transcription factors and translocation into the nucleus [111]. Studies have shown that HIV-1 protein nef is capable of modulating the TCR activation signals to promote HIV replication and pathogenesis by increasing the expression of the TCR-induced transcription factors NF-kB, NFAT, and interleukin 2 (IL-2) in both quiescent and metabolically active CD4+ T cells [162,163,164]. This pool of latent cells in the blood is relevant to HAND as it could allow re-entry to the CNS even if the virus is eliminated in the brain.

## 7. How to Address Neuropathogenesis and CNS Latent Reservoir

### 7.1. Shock and Kill Approach

The main hurdle to preventing HAND is the latent reservoir in microglial cells and perivascular macrophages [165]. One of the most popular strategies being studied currently to eliminate the latent reservoir, not just in the brain, is the “shock and kill” approach. The “shock and kill” approach entails reactivating the latent reservoir by the use of latency-reactivating agents (LRAs). Using LRAs, the provirus in latent cells would be expressed, which would either trigger the cells to die by a cytopathic effect or recognition by other immune cells such as cytotoxic T lymphocytes [166]. Unfortunately, this strategy has several obstacles and limitations when applied to the CNS. Two of the major challenges are—(i) addressing the heterogenous nature of the latent reservoir and (ii) controlling the damage caused by reactivation. For now, addressing the diverse latent pool is more important for researchers than determining how to prevent damage during reactivation. To address this, a variety of broad-acting drug classes are being studied [167]. One drug class developed includes those acting on epigenetic markers around the provirus, which include histone methyltransferase inhibitors (HMTis) and histone deacetyl transferase inhibitors (HDACis). This mechanism of reactivation would work by relaxing the chromatin to promote transcription factors to enable proviral replication [166]. Other drugs aim to target intracellular and paracellular signaling pathways. Some of these work by enhancing protein kinases in the JAK-STAT pathway, inhibiting the degradation of NF-κB or directly stimulating cytokine receptors [168,169]. Unfortunately, these drugs, even in synergistic combinations, have been insufficient to cause enough reactivation to make the “shock and kill” a viable strategy for the time being [170]. A further challenge to the “shock and kill” approach is minimizing damage in the CNS during reactivation. The cell death in the CNS during “shock and kill” would likely be severe, as the latently infected cells would succumb to death, but possibly uninfected cells could perish by a bystander effect. It is unknown how this cell death could alter the brain and what the long-term consequences could be to the central nervous system. 

### 7.2. Block and Lock Approach

On the flip side, another strategy termed the “block and lock” approach has gained traction in recent years to target the latent reservoir. This strategy aims to lock the latent reservoir into a “deep latency” wherein there is no production of the virus and very minimal to no transcription of the provirus. This would not eliminate the viral reservoir and confer sterilizing a cure but would lower intermittent reactivation and could potentially reduce neuroinflammation associated with viral shedding [171]. There have been a variety of compounds analyzed in the “block and lock” approach, but three classes that could see strong efficacy with regards to HAND are Tat inhibitors, LEDGINs, and NF-κB inhibitors.

As previously discussed, Tat plays a crucial role in the transcription of later viral transcripts as well as propagating neuroinflammation [19,21,22,24,25]. One particular Tat inhibitor of interest is didehydro-cortistatin A (dCA). dCA was shown to reduce the transcription of HIV and reactivation of latent cell lines and primary cells in vitro, as well as reducing reactivation in a mouse latency model [172,173,174]. dCA was also shown to create a restrictive epigenetic landscape around the HIV promoter by recruiting BAF and increasing the presence of deacetylated histone 3 at Nuc-1 [175].

LEDGINs are inhibitors of HIV integrase and have been shown to be effective in reducing HIV replication in the early and late stages of the cellular cycle [176,177]. Specifically, LEDGINs restrict early replication by allosterically inhibiting integrase. LEDGINs can also affect later stages of replication by promoting the formation of oligomers of integrase that result in the production of defective virions [176,178]. The early administration of LEDGINs could help to prevent the establishment of CNS latent reservoirs, and later administration could serve the “block and lock” approach to achieve a functional cure.

NF-кB inhibition is another path being investigated for the “block and lock” strategy. NF-кB was first shown to increase HIV replication around 50-fold back in 1987 [179]. More recently, there has been attention on inhibiting HSP90, which has been shown to regulate NF-кB, STAT5, and NFAT in a Tat-mediated manner [180,181]. One HSP90 inhibitor, AUY922, in combination with a reverse transcriptase inhibitor, EFdA, was even shown to prevent the reactivation of latent cells 11 weeks after administration in a mouse model [182]. Our group has been interested in sulforaphane (SFN), an isothyonate found in cruciferous vegetables, which has also been shown as an anticarcinogen [183,184]. We recently showed that SFN treatment can reduce the TNF-α-mediated reactivation of latently infected cells by attenuating NF-кB signaling by promoting the antioxidant NRF2 [185]. SFN remains of interest for the CNS as its antioxidant properties can help prevent disease progression associated with oxidative stress in HAND [186]. SFN has also been shown to disrupt the activation of the NLRP3 inflammasome, which could majorly reduce IL-1ß and TNF-α signaling mediated by microglia during HIV infection [187]. Our group is continuing to investigate how the dual anti-HIV and antioxidant effects of SFN could be used to combat HAND (Figure 1). With this said, NF-кB signaling remains one of the top pathways of interest to inhibit in order to promote the “block and lock approach”.

### 7.3. Alternate Treatment and Concluding Remarks

Currently, the best approach to preventing HAND is through cART. cART helps to prevent the progression of ANI to MNI or HAD. Improving the CD4+ T cell count and suppressing viral load through cART has been shown to alleviate neurocognitive impairments [188]. Whether addressing cART, “shock and kill”, or “block and lock”, one large hurdle to overcome is getting the drugs to cross the BBB to travel to the microglia while avoiding neurotoxicity. Currently, antiretrovirals with a greater ability to permeate the BBB have been shown to reduce viral load but have also been shown to have a greater degree of neurotoxic effects and patients who cease taking them have even been shown to have an increase in neurocognitive functioning [189,190]. This creates a very challenging double-edged sword for treatment and management with no clear solution currently. For most patients on cART, the main symptoms experienced are mood disorders. Treatment for these symptoms has been successful with standard antidepressant drug classes like serotonin reuptake inhibitors [191,192]. A 2018 study found that a CCR2 and CCR5 inhibitor, cenicriviroc, was able to improve neurocognitive impairments in patients with cART [193]. In conclusion, while researchers have improved the treatment of HAND over the past 35 years, until the latent reservoir can be fully purged or driven to a highly non-productive state, we will continue to need cART and develop new drug regimens to improve and prevent neurocognitive impairments.

## Figures and Tables

**Figure 1 ijms-25-04697-f001:**
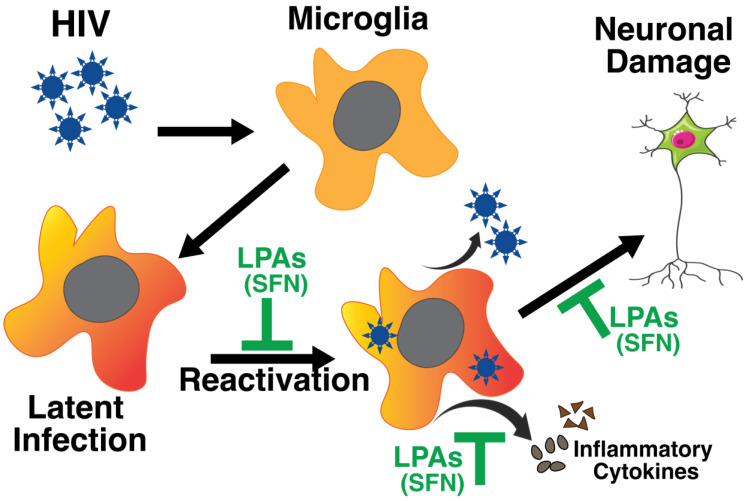
Schematic diagram depicting that microglial cells can be infected by HIV and establish a latent reservoir. However, this reservoir is subject to intermittent reactivation that produces virus particles, viral proteins, and inflammatory cytokines. Neurotoxic viral proteins and cytokines cause neuronal damage and contribute to progressive neurodegenerative consequences. The identification of novel latency promoting agents such as SFN is highly desirable as it can prevent the reactivation of HIV as well as inhibit cytokine release and thus may prevent HIV-associated neuropathology.

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
