# Peer review of "HIV-Associated Neurocognitive Disorder: A Look into Cellular and Molecular Pathology"

_ijms, 2024, doi:10.3390/ijms25094697_

Round 1

Reviewer 1 Report

Comments and Suggestions for Authors

HIV Associated Neurocognitive Disorder: A Look into Cellular and Molecular Pathology

I found this is an excellent compilation of a review article on HIV Associated Neurocognitive Disorder (HAND). The review article provides updates, describes the mechanism and challenges associated with the onset and progression of HAND, and further discusses current management strategies. This article is meticulously formulated and well-written and will help boost current mechanistic updates on HAND among readers. Following are the specific comments to further strengthen the present manuscript,

1.  In the introduction, line no. 35. Lastly, ANI, please correct this; I assume this will be HAD.

2.  In line 46, the sentence can be modified to “within 15 days of exposure, HIV enters the CNS.

3.  PMC should be mentioned in abbreviated form in line 65, next to platelet monocyte complexes. Suddenly, PMC appears in line 68.

4.    In the line 249, a reference may be cited.

5.    Line 287, 288: Please rewrite the sentence for better expression.

6.    Line 328, revisit partake and correct any typos.

7.  I detected a mismatch in citations; please revisit and correct all citations. For instance, in line no. 363 [84, 85] appeared to me as needing to change to [90, 91]; the statements are about Tat and gp120 mediated activation of NLRP3 inflammasome. This issue must be fixed.

8.    Also, in line 366, citation [86] needs to be corrected; please consider redoing the citation and matching carefully.

9.  Line 394, integrate, consider rewriting the entire sentence for better understanding.

Author Response

I found this is an excellent compilation of a review article on HIV Associated Neurocognitive Disorder (HAND). The review article provides updates, describes the mechanism and challenges associated with the onset and progression of HAND, and further discusses current management strategies. This article is meticulously formulated and well-written and will help boost current mechanistic updates on HAND among readers. Following are the specific comments to further strengthen the present manuscript,

Response: We would like to thank the reviewer for positive comments and feedback. We thoroughly edited the manuscript for errors and consistency. We have also included an acknowledgement section to thank funding sources and other lab members who assisted with editing and proofreading of the revised manuscript.  Please find our point-by-point response below-

  1. In the introduction, line no. 35. Lastly, ANI, please correct this; I assume this will be HAD.

Response: We have corrected it.

  1. In line 46, the sentence can be modified to “within 15 days of exposure, HIV enters the CNS.

Response: We have made the change. Thank you!

  1. PMC should be mentioned in abbreviated form in line 65, next to platelet monocyte complexes. Suddenly, PMC appears in line 68.

Response: It has been fixed.

  1. In the line 249, a reference may be cited.

Response: We have added relevant references.

  1. Line 287, 288: Please rewrite the sentence for better expression.

Response: We have modified the sentence and hope it expresses the information clearly.

  1. Line 328, revisit partake and correct any typos.

Response: It has been fixed.

  1. I detected a mismatch in citations; please revisit and correct all citations. For instance, in line no. 363 [84, 85] appeared to me as needing to change to [90, 91]; the statements are about Tat and gp120 mediated activation of NLRP3 inflammasome. This issue must be fixed.

Response: We would like to thank the reviewer for catching this oversight. We have carefully checked the references for their relevance and made changes where they were needed.

  1. Also, in line 366, citation [86] needs to be corrected; please consider redoing the citation and matching carefully.

Response: It has been corrected.

  1. Line 394, integrate, consider rewriting the entire sentence for better understanding.

Response: We have written the sentence for better understanding. Thanks!

Reviewer 2 Report

Comments and Suggestions for Authors

The authors present an exciting and complete review article discussing the cellular and molecular changes occurring during the neurocognitive disorders associated with HIV. The language used is correct, and only two subsections will be needed to complement the information provided. I advise the authors to read and mention the references provided.

-The authors should write a small subsection discussing the risk factors that can exacerbate these neurocognitive disorders. This is an excellent reference (DOI: 10.1007/s11904-014-0210-3). The authors should mention that the cytokine/chemokine response and changes in the immune system happening during co-infections have the potential to aggravate the symptoms of these neurocognitive disorders (DOI: 10.1016/j.heliyon.2023.e15055; 10.1038/nrmicro.2017.128).

-The authors should review the relevant literature and discuss the biomarkers used to evaluate HAND (viral markers, host markers, markers in the cerebrospinal fluid, and limitations of the markers). It would be an excellent short section describing or mentioning those macromolecules used today as biomarkers of HAND.

Author Response

The authors present an exciting and complete review article discussing the cellular and molecular changes occurring during the neurocognitive disorders associated with HIV. The language used is correct, and only two subsections will be needed to complement the information provided. I advise the authors to read and mention the references provided.

Response: We would like to thank the reviewer for very insightful comments and suggestions for the addition of very relevant information. We have realized that the addition of this information was appropriately warranted and has made our review article more comprehensive.

-The authors should write a small subsection discussing the risk factors that can exacerbate these neurocognitive disorders. This is an excellent reference (DOI: 10.1007/s11904-014-0210-3). The authors should mention that the cytokine/chemokine response and changes in the immune system happening during co-infections have the potential to aggravate the symptoms of these neurocognitive disorders (DOI: 10.1016/j.heliyon.2023.e15055; 10.1038/nrmicro.2017.128).

 Response: This subsection has been added as subsection 5.3.

-The authors should review the relevant literature and discuss the biomarkers used to evaluate HAND (viral markers, host markers, markers in the cerebrospinal fluid, and limitations of the markers). It would be an excellent short section describing or mentioning those macromolecules used today as biomarkers of HAND.

Response: This subsection has been added as subsection 4.4.